# Inhibitor of ppGalNAc-T3-mediated O-glycosylation blocks cancer cell invasiveness and lowers FGF23 levels

**Lina Song, Adam D Linstedt\***

Department of Biological Sciences, Carnegie Mellon University, Pittsburgh, United States

**Abstract** Small molecule inhibitors of site-specific O-glycosylation by the polypeptide N-acetylgalactosaminyltransferase (ppGalNAc-T) family are currently unavailable but hold promise as therapeutics, especially if selective against individual ppGalNAc-T isozymes. To identify a compound targeting the ppGalNAc-T3 isozyme, we screened libraries to find compounds that act on a cell-based fluorescence sensor of ppGalNAc-T3 but not on a sensor of ppGalNAc-T2. This identified a hit that subsequent in vitro analysis showed directly binds and inhibits purified ppGalNAc-T3 with no detectable activity against either ppGalNAc-T2 or ppGalNAc-T6. Remarkably, the inhibitor was active in two medically relevant contexts. In cell culture, it opposed increased cancer cell invasiveness driven by upregulated ppGalNAc-T3 suggesting the inhibitor might be anti-metastatic. In cells and mice, it blocked ppGalNAc-T3-mediated glycan-masking of FGF23 thereby increasing its cleavage, a possible treatment of chronic kidney disease. These findings establish a pharmacological approach for the ppGalNAc-transferase family and suggest that targeting specific ppGalNAc-transferases will yield new therapeutics.

**\*For correspondence:** linstedt@ andrew.cmu.edu

**Competing interests:** The authors declare that no competing interests exist.

## Introduction

Glycosylation, which fine-tunes the function of proteins, is the most abundant and diverse posttranslational modification (*Schjoldager and Clausen, 2012*; *van der Post et al., 2013*; *Moremen et al., 2012*). Despite many documented roles in health and disease, the enzymes that mediate mucin-type O-glycosylation in the Golgi apparatus have yet to be discovered as druggable targets. The initiating enzymes, a family of 20 ppGalNAc-transferase isozymes, determine which substrates are modified and at which sites. Significant questions remain regarding their specificity, regulation, targets and functions and the lack of a pharmacological approach has been a critical limitation. The only confirmed inhibitor of mucin-type O-glycosylation, benzyl-N-acetyl-$\alpha$-galactosaminide, blocks elongation rather than initiation and requires millimolar concentrations that can be toxic (*Kuan et al., 1989*; *Patsos et al., 2009*). Not only are pan-specific modulators of the ppGalNAc-transferases lacking, there is nothing isoform-specific. The latter will be essential to restrict effects to particular pathways and could possibly lead to a new basis for therapeutics conceptually related to the widespread use of drugs targeting individual protein kinases.

ppGalNAc-T3 serves as an important test case as it is implicated in at least two medically important pathways: cancer metastasis and stabilization of FGF23 (*Chefetz and Sprecher, 2009*; *Schjoldager et al., 2011*; *Kato et al., 2006*; *Peng et al., 2012*; *Kohsaki et al., 2000*; *Kitada et al., 2013*; *Gao et al., 2013*; *Brockhausen, 2006*; *Brooks et al., 2007*). ppGalNAc-T3 is overexpressed in cancerous tissue often correlating with shorter survival (*Kitada et al., 2013*; *Brockhausen, 2006*; *Harada et al., 2016*; *Mochizuki et al., 2013*). Knockdown of ppGalNAc-T3 expression in cultured ovarian cancer cells inhibits their invasive capacities arguing that ppGalNAc-T3 has potential as a

**eLife digest** Complex cascades of interactions between different molecules regulate every process in the body. Enzymes are critical for this, because they act as 'catalysts' to speed up chemical reactions in a cell. Each type of enzyme has a specific role. The enzyme ppGalNAc-T3, for example, attaches sugar molecules onto certain proteins via a process called glycosylation. This modification fine-tunes the activity of the proteins.

The enzyme ppGalNAc-T3 is implicated in at least two medically important pathways. It increases the amount of the hormone FGF23, which regulates phosphate levels in the bloodstream. Hormones are messenger molecules that regulate most processes that are crucial for life, and too much or too little of a hormone can lead to diseases. High levels of FGF23, for example, can cause serious and often fatal problems in patients with chronic kidney disease.

The enzyme ppGalNAc-T3 is also known to encourage and stimulate some cancer cells to spread from their original location to other parts of the body – a process known as metastasis – which ultimately leads to the vast majority of cancer deaths every year. Thus, a drug or molecule that blocks ppGalNAc-T3 could be used to lower FGF23 levels in patients with kidney disease and potentially prevent cancer cells from spreading. However, until now, it was unknown if such a molecule existed.

To identify a compound that specifically regulates ppGalNAc-T3, Song and Linstedt engineered human cells grown in the laboratory to become fluorescent when ppGalNAc-T3 was blocked. Out of the over 20,000 compounds screened, one compound named T3Inh-1 selectively blocked ppGalNAc-T3. Further experiments showed that the T3Inh-1 compound reduced FGF23 hormone levels in both tissue cells grown in the laboratory and mice, without causing any toxic side effects. It also prevented breast cancer cells grown in the laboratory from spreading. The results demonstrated that T3Inh-1 is the first drug-like inhibitor that can target this kind of enzyme.

An important next step will be to test the compound in animal models for chronic kidney disease and cancer metastasis.

therapeutic target (*Wang et al., 2014*). ppGalNAc-T3 mediates glycan-masking of FGF23 in bone as part of a control mechanism determining the form of FGF23 that is secreted (*Kato et al., 2006*; *Tagliabracci et al., 2014*). When present, the added O-glycan blocks FGF23 cleavage by the furin protease resulting in secretion of intact FGF23 that activates FGF23 receptor complexes at the kidney and intestine (*Rowe, 2015*). In contrast, non-glycosylated FGF23 is cleaved and the cleaved C-terminal product competitively blocks these same receptors (*Goetz et al., 2010*). Significantly, elevated intact FGF23 occurs in chronic kidney disease and upon kidney transplant where it is directly linked to poor prognosis due to its effects on renal phosphate reabsorption and 1,25-dihydroxyvitamin D biosynthesis (*Eckardt and Kasiske, 2009*). Many studies have concluded that therapeutic control over FGF23 would be transformative in the clinic (*Degirolamo et al., 2016*; *Fukumoto, 2016*; *Isakova et al., 2015*; *Smith, 2014*).

As a step toward testing whether biologic discoveries and new therapeutics will result from developing small molecule modulators that control the activity of specific ppGalNAc-transferases we initiated high throughput cell-based screening of compound libraries. An inhibitor of ppGalNAc-T3 was identified and found to block breast cancer cell invasiveness as well as secretion of intact FGF23 promising new therapeutic approaches for cancer and chronic kidney disease, respectively.

## Results and discussion

HEK cell lines were engineered to express fluorescent sensors that are specific to ppGalNAc-T2 or ppGalNAc-T3 activity (*Song et al., 2014*). For each sensor, glycosylation of its isozyme-specific target site prevents furin protease from removing a blocking domain (*Figure 1A*). Thus, fluorescence increases upon ppGalNAc-transferase inhibition because removal of the blocking domain allows dimerization of a fluorogen activating protein domain so that it binds and activates the fluorescence of malachite green. They are ratiometric because the sensor backbone contains a green fluorescent protein as an internal control for expression. Each sensor showed clear activation after mutation of

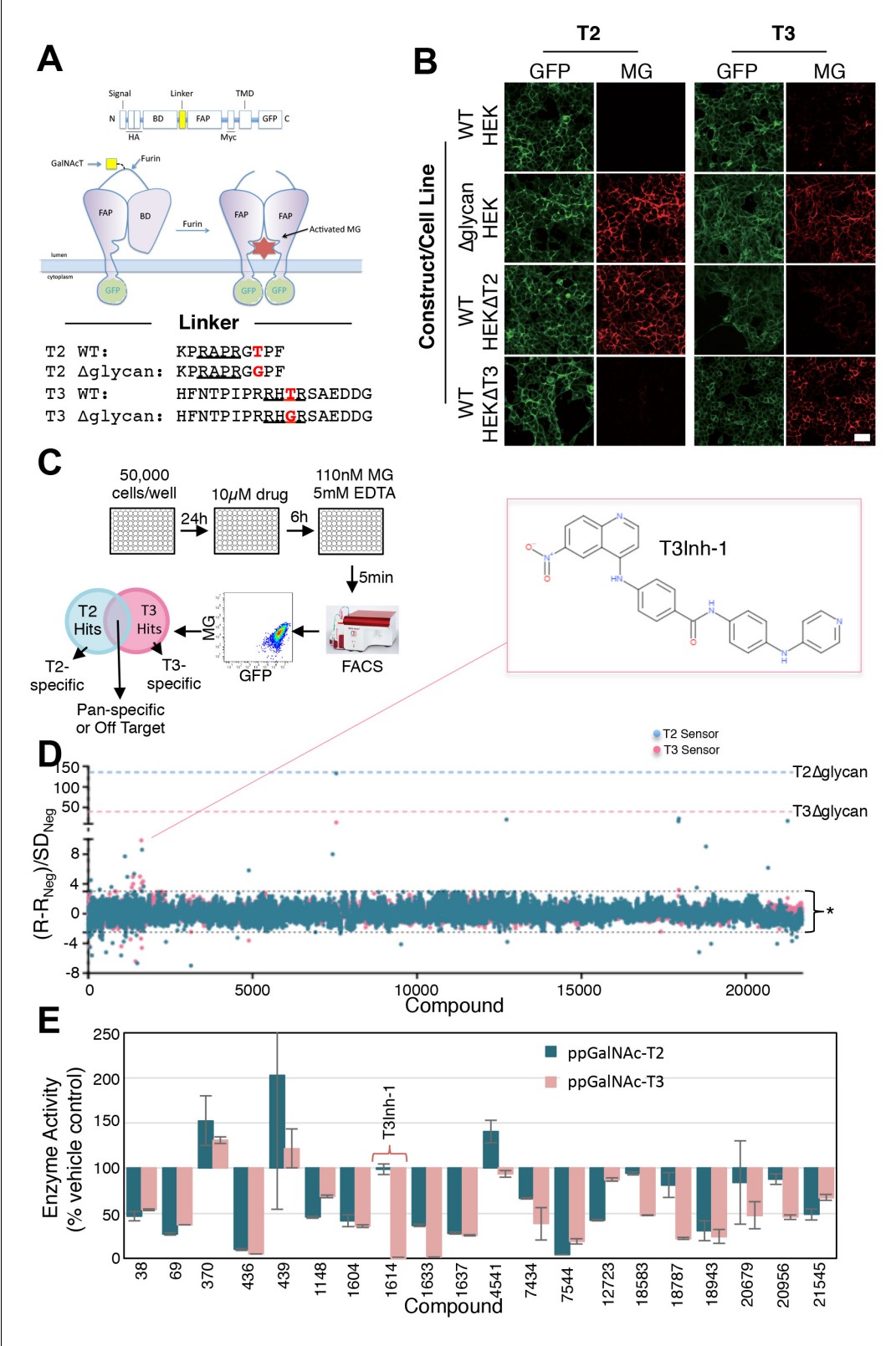

**Figure 1.** Screen for modulators of ppGalNAc-T2/T3. (**A**) Diagram showing sensor design and the linker sequences used. O-glycosylation of the linker masks the furin site but if an inhibitor blocks the ppGalNAc-transferase then furin cleaves the linker releasing the blocking domain (BD) allowing fluorescent activating protein (FAP) dimerization and dye activation. Linker furin sites are underlined and sites of glycosylation or mutation are in bold. (**B**) HEK cell lines with or without ppGalNAc-T2 or T3 stably expressing the WT or Δglycan T2 or T3 sensor constructs (see linkers in A) were imaged in

*Figure 1 continued on next page*

*Figure 1 continued*

the presence of 110 nM of the dye MG11p (MG) to detect GFP or MG. Bar = 20 µm. (**C**) Schematic showing cell plating, drug treatment, cell release, fluorescence measurement and parallel analysis using both T2 and T3 sensors. Hits that activate both may be pan-specific or act on off-target pathways common to both sensors whereas sensor specific hits are likely acting directly on the corresponding ppGalNAc-transferase. (**D**) The plot shows Q values (Q=(R-R$_{Neg}$)/SD$_{Neg}$) for each compound (treatment at 10 µM for 6 hr) using the average of duplicate MG/GFP ratios for the compound (R), the vehicle-only control (R$_{Neg}$), and the standard deviation of the vehicle-only controls (SD$_{Neg}$). The cut-off values of +3 and −2.5 are indicated (*). Also indicated are the values for the positive controls (T2Δglycan and T3Δglycan) and the structure of the indicated T3-specific hit (inset). (**E**) Values (% enzyme activity relative to vehicle-only controls) in the in vitro assay using purified ppGalNAc-T2 or T3 as a secondary screen are shown for 20 hits from the primary screen. Compounds were present at 50 µM. Compound 1614 is T3Inh-1.

The following source data and figure supplements are available for figure 1:

**Source data 1.** Primary screen data for HEK cells expressing T2 or T3 sensors.

**Source data 2.** Secondary screen data (in vitro enzyme assays).

**Figure supplement 1.** Cell growth at various T3Inh-1 exposures.

**Figure supplement 1—source data 1.** Cell counts at differing time points and T3Inh-1 concentrations.

**Figure supplement 2.** General N- and O-glycosylation are unaffected.

**Figure supplement 2—source data 1.** Fluorescent lectin staining of cells at differing T3Inh-1 concentrations.

**Figure supplement 3.** ppGalNAc-transferase levels are unaffected.

its glycan acceptor sites and these mutated constructs served as positive controls in the screen (*Figure 1B*, Δglycan). The T3 sensor exhibited a background level of activation due to incomplete glycosylation (*Song et al., 2014*) but this was considered advantageous for the possible identification of enzyme activators along with the desired inhibitors. Consistent with our previous report (*Song et al., 2014*), sensor expression in HEK cell lines depleted of either ppGalNAc-T2 or T3 via zinc finger nuclease editing resulted in specific activation of the corresponding sensor confirming their isozyme selectivity (*Figure 1B*, HEKΔT2, HEKΔT3). Our screen included compounds based on structural diversity (21,710 compounds in total) with 6 hr treatments at 10 µM prior to flow cytometry to assay MG and GFP fluorescence on a cell-by-cell basis (*Figure 1C*). Each compound was tested in duplicate and against both sensors (*Figure 1—source data 1*). Because each sensor requires essentially identical cellular reactions- the only difference being which ppGalNAc-transferase isoform modifies the sensor- most off-target hits (such as sugar nucleotide transporters, extending enzymes, or furin) will alter both sensors, whereas directly acting, isoform-specific candidates will be sensor-specific. Using cut-off parameters for the MG/GFP ratios that excluded >99% of the compounds (Q ≥ 3 or Q ≤ −2.5), the screen yielded 72 sensor-specific hits with 18 increasing and 35 decreasing the T2 sensor fluorescence and 11 increasing and 8 decreasing the T3 sensor fluorescence (*Figure 1D*).

To determine which of these directly acted on the targeted enzymes, we carried out in vitro glycosylation assays in which the purified lumenal domains of ppGalNAc-T2 or ppGalNAc-T3 (containing catalytic and lectin domains) were incubated with peptide and UDP-GalNAc substrates in the presence of 50 µM of each compound. A second stage reaction (UDP-Glo) then converted the accumulated UDP product to ATP and then, via luciferase, to light. This resulted in 20 candidates that either reduced or increased the luminescence by a factor ≥50% relative to vehicle-only controls (*Figure 1E*). Of these, one compound (#1614) stood out as a strong and selective inhibitor of ppGalNAc-T3 and became the focus of this phase of the study. The compound is a quinoline of no known activity that we now refer to as ppGalNAc-T3 Inhibitor 1 or T3Inh-1 (*Figure 1D*, inset).

Importantly, T3Inh-1 exhibited no toxicity and did not affect HEK cell proliferation (*Figure 1—figure supplement 1*). Also, a 24 hr treatment did not affect staining intensity by the lectins Concanavalin A (ConA), Wheat germ agglutinin (WGA), Sambucus Nigra (SNA) or Vicia Villosa (VVA), which

bind branched alpha-mannose, N-acetylglucosamine, sialic acid, or terminal GalNAc, respectively (*Figure 1—figure supplement 2*). This implies that the enzymes contributing to the abundant glycans of N- and O-glycosylation detected by these lectins were unaffected and is consistent with ppGalNAc-T3 modifying a relatively limited number of substrates (*Schjoldager et al., 2015*). Finally, there was no change in localization or expression level of ppGalNAc-T3, ppGalNAc-T2 or any other Golgi marker tested (*Figure 1—figure supplement 3*).

To determine the effective concentration of T3Inh-1 in cells and in vitro, it was retested at various doses against the sensors and against the purified enzymes. T3Inh-1 activated the T3 sensor with an apparent $IC_{50}$ of 12 μM and showed little or no activity towards the T2 sensor (*Figure 2A*). Similarly, T3Inh-1 was a potent and selective direct inhibitor of ppGalNAc-T3 (*Figure 2B*). Inhibition of ppGal-NAc-T3 occurred with an $IC_{50}$ of 7 μM and was undetectable against ppGalNAc-T2. T3Inh-1 also lacked activity against ppGalNAc-T6 (*Figure 2B*), which is the isozyme considered most closely related to ppGalNAc-T3 (*Bennett et al., 2012*; *Revoredo et al., 2016*).

Towards characterizing the mechanism of inhibition we used the in vitro assay with purified ppGalNAc-T3 and individually varied both peptide and UDP-GalNAc substrate concentrations in the presence of 0, 7.5 or 15 μM T3Inh-1. The results were similar for both substrates (*Figure 2C–D*), where T3Inh-1 decreased the Vmax and increased the Km (Table 1) indicating a mixed-mode of inhibition in which the inhibitor most likely binds both free enzyme to reduce substrate binding and enzyme-substrate complexes to reduce turnover. Implied in this model of action is direct binding to the enzyme typically at an allosteric site. To test for direct binding, intrinsic tryptophan fluorescence of ppGalNAc-T3 was determined in the presence of increasing T3Inh-1 concentrations. At all concentrations, the compound itself yielded miniscule signals, whereas the compound had a profound and dose-dependent effect on the ppGalNAc-T3 emission spectrum (*Figure 2E*). These results confirmed direct binding with an apparent Kd of 17 μM (*Figure 1F*). The similarity in concentration dependence of sensor activation in cells, in vitro inhibition and direct binding argues that T3Inh-1 acts directly on cellular ppGalNAc-T3 and inhibits its activity.

Given its validation as a direct inhibitor of ppGalNAc-T3 without obvious off-target effects we turned to biologically relevant tests of T3Inh-1. As mentioned, overexpression of ppGalNAc-T3 is linked to cancer cell invasiveness as well as poor outcomes in patients (*Kitada et al., 2013*; *Brockhausen, 2006*; *Harada et al., 2016*; *Mochizuki et al., 2013*). Although no linkage to breast cancer has been reported, our analysis of publically available data for 1117 breast cancer patients (*Szász et al., 2016*) using Kaplan-Meier survival plots shows that high expression of ppGalNAc-T3 correlates with poor patient overall and metastasis-free survival (*Figure 3—figure supplement 1A–B*). Therefore, we carried out migration and invasion assays with the breast cancer cell line MDA-MB231, which expresses a relatively high level of ppGalNAc-T3 (*Figure 3—figure supplement 1C*), in the absence or presence of 5 μM T3Inh-1. Cells were cultured on uncoated (to assay migration) or Matrigel-coated (to assay invasiveness) Bioboat filters for 24 or 48 hr and those cells that moved to the underside of the filters were imaged and quantified. T3Inh-1 was strikingly effective, inhibiting migration by >80% (*Figure 3A*) and invasion by 98% (*Figure 3B*) while causing no discernable effect on cell proliferation (*Figure 3C*). To confirm that the effect was due to ppGalNAc-T3, the same experiment was carried out using MCF7 cells, which is a breast cancer cell line that expresses relatively low levels of ppGalNAc-T3 (*Figure 3—figure supplement 1C*). Critically, invasion by MCF7 cells was significantly increased by transfection with ppGalNAc-T3 and this increase was strongly blocked by T3Inh-1 (*Figure 3D*). Although O-glycosylation has been connected to cancer migration and invasion there is little evidence that it could be targetable for clinical purposes. Our results provide a 'proof of concept' and a strong starting point for developing the necessary tools.

As the relevant target(s) of ppGalNAc-T3 that drive metastatic-like cell behavior remain to be identified, we next tested whether T3Inh-1 could inhibit glycan masking of FGF23, a known ppGal-NAc-T3 target. If so, we expected reduced secretion of intact FGF23. HEK cells co-expressing transfected FGF23 and ppGalNAc-T3 were treated with increasing concentrations of T3Inh-1 and secreted FGF23 was assayed by immunoblot. There was a clear dose-dependent loss of intact FGF23 (*Figure 4A*) and an increase in the ratio of cleaved/intact yielding a half-max of 14 μM for this effect (*Figure 4B*). For an unknown reason, perhaps related to its instability in media (*Kato et al., 2006*), the cleaved fragment did not show a corresponding increase. Rather, its recovery varied with the average over three experiments yielding a relatively small increase (*Figure 4—figure supplement 1A–B*). As expected, intact and cleaved FGF23 showed no change in cell lysates even for cells

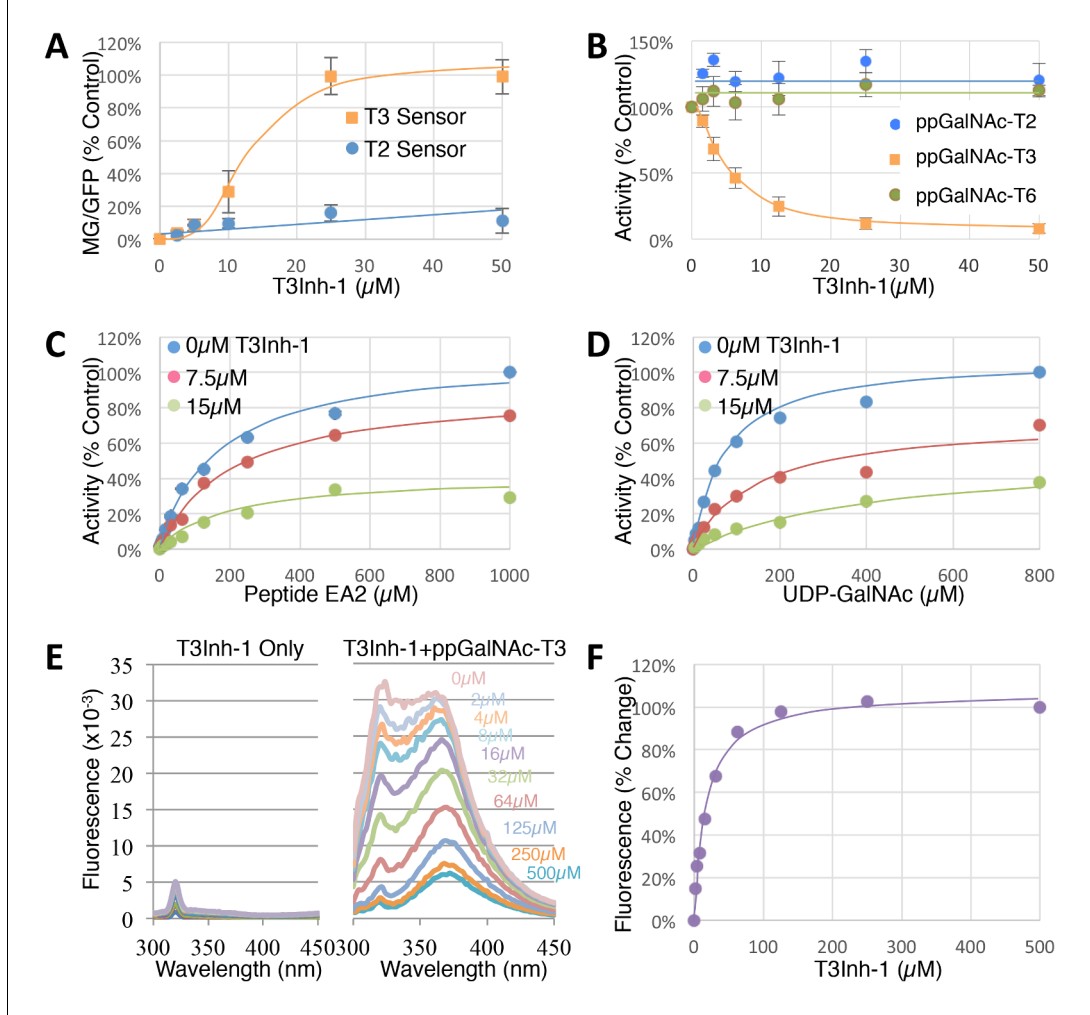

**Figure 2.** T3Inh-1 is a direct mixed-mode inhibitor of ppGalNAc-T3. (**A**) Comparison of T2 and T3 sensor activation at the indicated concentrations of T3Inh-1 (n = 3 ± SEM). MG/GFP ratio was determined for 20,000 cells by FACS and average value is plotted as percent of the positive control (i.e. the Δglycan version of each sensor). (**B**) Comparison of effect of the indicated concentrations of T3Inh-1 on in vitro glycosylation mediated by purified ppGalNAc-T2, ppGalNAc-T3, or ppGalNAc-T6. Values are averages expressed as percentage of the control 'vehicle-only' reactions (n = 6 ± SEM for ppGalNAc-T3, n = 3 ± SEM for others). (**C–D**) The in vitro assay was carried out in the presence of 0, 7.5, or 15 µM T3Inh-1 at the indicated concentrations of peptide or UDP-GalNAc substrate. Values are averages expressed as percent of the control reactions with no inhibitor and saturating substrates (n = 3 ± SEM). (**E**) Representative fluorescence spectra are shown for T3Inh-1 alone or for purified ppGalNAc-T3 in the presence of the indicated concentrations of T3Inh-1. Note dose-dependent quenching of tryptophan fluorescence indicating direct binding. (**F**) Fluorescence quenching was quantified at each concentration using the peak value at 324 nm (n = 3 ± SEM). Note that all graphs have error bars but some are too small to be apparent.

The following source data is available for figure 2:

**Source data 1.** Panel A: sensor signals versus T3Inh-1 concentration.

**Source data 2.** Panel B: enzyme activity versus T3Inh-1 concentration.

**Source data 3.** Panel C: inhibitor effect versus peptide concentration.

**Source data 4.** Panel D: inhibitor effect versus UDP-GalNAc concentration.

**Source data 5.** Panel E and F: Fluorescence change versus T3Inh-1 concentration.

**Table 1.** Inhibition by T3Inh-1 at varying substrate concentrations. Values shown were determined from the data in **Figure 2** using Prism (see Materials and methods).

| Substrate | Parameter | 0 μM | 7.5 μM | 15 μM |
|---|---|---|---|---|
| Peptide (EA2) | Vmax | 100% | 82% | 36% |
| | Km (μM) | 173.7 | 208.4 | 210.3 |
| | Ki (μM) | 9.9 | | |
| UDP-GalNAc | Vmax | 100% | 71% | 56% |
| | Km (μM) | 74.9 | 153.4 | 448.3 |
| | Ki (μM) | 2.9 | | |

with lysosomal degradation inhibited by chloroquine (*Figure 4—figure supplement 1C–E*) arguing that T3Inh-1 affected cleavage just prior to secretion and did not affect FGF23 expression or cause intracellular routing to lysosomes. Importantly, we also tested the effect of T3Inh-1 on cleavage of ANGPTL3, which is controlled by ppGalNAc-T2-mediated glycan masking (*Schjoldager et al., 2012*, *2010*). Secreted intact ANGPTL3 remained high at all concentrations, confirming the selectivity of T3Inh-1 towards ppGalNAc-T3 (*Figure 4A–B*). Reasoning that we might be able to see a similar effect on secreted FGF23 in an animal model, mice were injected intraperitoneally with T3Inh-1 and serum levels of cleaved FGF23 were determined. Three groups (0, 25 and 50 mg/kg T3Inh-1) of mice received either one or two injections separated by 24 hr, followed by blood collection after another 24 hr. There were no apparent ill effects on animal health. An ELISA assay with antibodies against the N- and C-terminal portions of FGF23 was used to determine the ratio of cleaved/intact FGF23 in the blood (*Sun et al., 2015*; *Bai et al., 2016*). Remarkably, T3Inh-1 caused a robust and statistically significant increase in this ratio at the tested 25 and 50 mg/kg concentrations (*Figure 4C*). These findings support the further development of T3Inh-1 toward mitigating the effects of elevated FGF23 signaling in chronic kidney disease patients.

This study identifies an isozyme-selective inhibitor targeting ppGalNAc-T3. The compound binds directly conferring a mixed-mode of inhibition and is equally active in vitro and in cells. Its discovery paves the way for structural studies that will contribute to our understanding of the enzyme reaction mechanism and guide rational design of modified versions of T3Inh-1 to improve its binding affinity and efficacy. Forthcoming tests of disease models, possibly employing higher affinity versions, may strengthen the case for therapeutic uses of T3Inh-1. It is difficult to predict possible side effects because a full list of ppGalNAc-T3 substrates is not yet available. However, the known effects of its knockout in the mouse model are all attributed to FGF23 processing (*Ichikawa et al., 2014*). Thus, use of T3Inh-1 to reduce intact FGF23 (and increase the inhibitory cleaved fragment) to treat chronic kidney disease may have limited side effects. Clearly, the issue of multiple substrates has not been a major concern in successful therapies targeting protein kinases. To conclude, we anticipate rational design aided by T3Inh-1, as well as further screening using isozyme-specific sensors, to result in a panel of both isozyme- and pan-specific modulators targeting ppGalNAc-transferases. As individual ppGalNAc-transferase isozymes are associated with unique diseases (*Schjoldager and Clausen, 2012*; *van der Post et al., 2013*; *Moremen et al., 2012*), the result would be a new class of therapeutics capable of treating an array of differing diseases.

## Materials and methods

### Cell lines and antibodies

HEK cell lines were previously described (*Song et al., 2014*). HeLa (Cat#ATCC-CCL-2, CVCL_0030), MDA-MB231 (Cat#ATCC-HTB-26, CVCL_0062), and MCF7 (Cat#ATCC-HTB-22, CVCL_0031) were purchased from ATCC (Manassas, VA). All cell lines were verified mycoplasma free every two months using Hoechst staining. Antibodies used were monoclonal antibodies 4C4 against ppGalNAc-T2 and UH5 against ppGalNAc-T3 (8, 31, 36), monoclonal 9e10 against the myc epitope (*Evan et al., 1985*; *Jesch et al., 2001*), a polyclonal against the FLAG epitope (Bethyl Labs, Cat#A190-102B, AB_

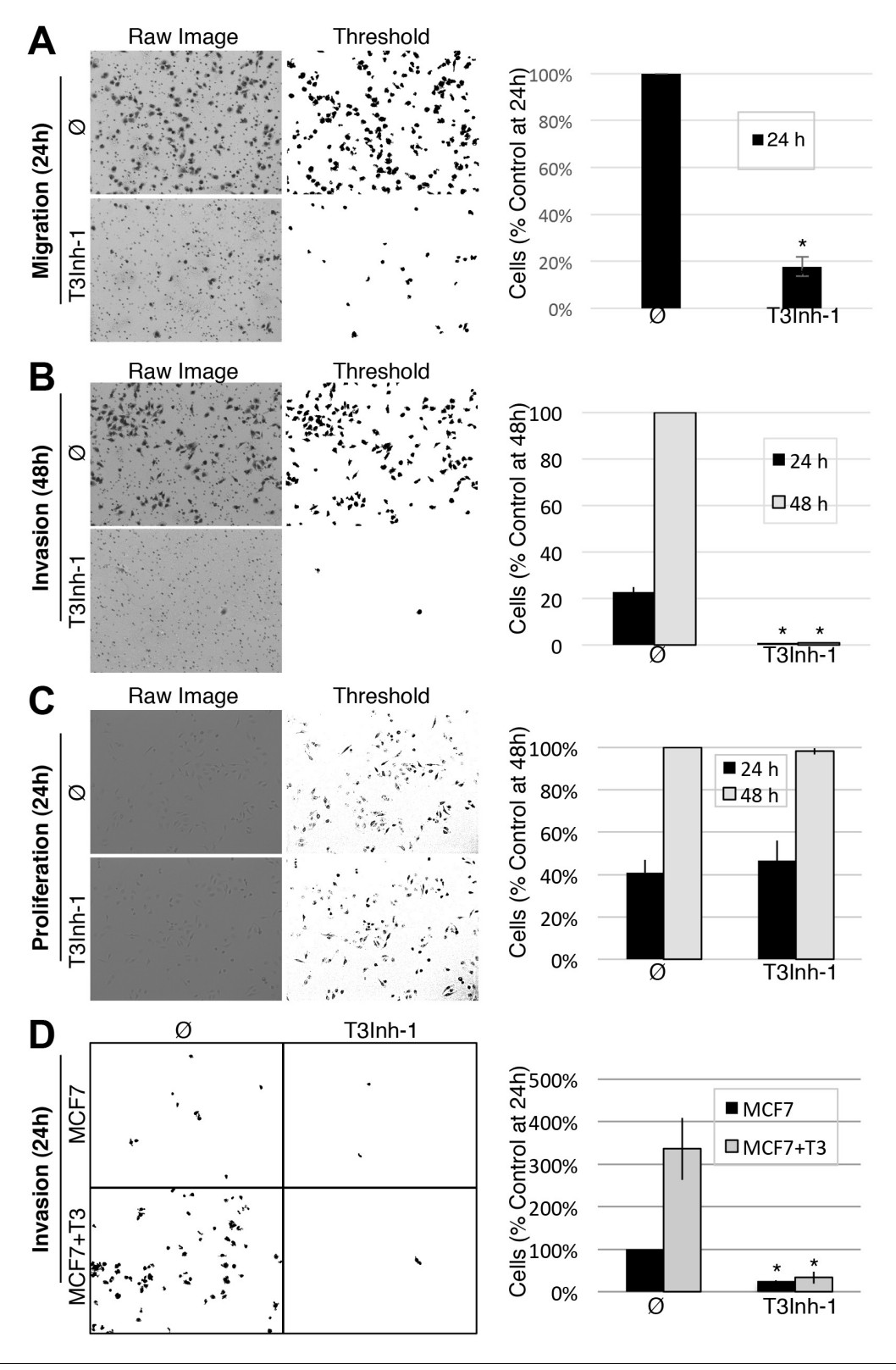

**Figure 3.** T3Inh-1 inhibits cell invasion. (**A**) Cell migration through uncoated filters was determined for the MDA-MB231 breast cancer cell line grown in the absence or presence of 5 µM T3Inh-1. The raw image of the filter shows both cells and the filter holes whereas a size-cut off was used in the thresholded image to specifically visualize the cells. Results were quantified by counting cells that migrated to the underside of the filter and each

*Figure 3 continued on next page*

*Figure 3 continued*

experiment was normalized using the average determined for controls at 24 hr (n = 3 ± SEM). (**B**) Identical analysis except that the filters were pre-coated with Matrigel so that the assay measures invasion not just migration and the 48 hr control was used for normalization (n = 3 ± SEM). (**C**) MDA-MB231 proliferation was determined for cells grown in the presence or absence of 5 µM T3Inh-1 by cell counting at 24 or 48 hr. Representative images before and after thresholding (no size cutoff) are shown as well as quantification normalized by the value determined for untreated cells at 48 hr (n = 3 ± SEM). (**D**) Mock and ppGalNAc-T3 transfected MCF7 cells were plated on Matrigel-coated filters in the absence or presence of 5 µM T3Inh-1 for 24 hr. Thresholded images show cells on underside of filters. Cell counts are shown relative to untreated controls after normalization using the total number of cells (determined using parallel wells 24 hr post-plating). For all panels, asterisks denote p<0.05 (two-tailed Student's t test) for untreated to T3Inh-1 comparison.

The following source data and figure supplement are available for figure 3:

**Source data 1.** Panel A: Cell counts in migration assay.

**Source data 2.** Panel B: Cell counts in invasion assay.

**Source data 3.** Panel C: Cell counts in proliferation assay.

**Source data 4.** Panel D: Cell counts in MCF7 invasion assay.

**Figure supplement 1.** Breast cancer survival as a function of ppGalNAc-T3 expression and ppGalNAc-T3 expression in cultured breast cancer cell lines.

1944186), a polyclonal against GPP130 (*Puri et al., 2002*), a purchased anti-ppGalNAc-T3 antibody (ThermoFisher, Cat#PA5-25217, AB_2542717), an anti-α-tubulin antibody (Biolegend, Clone TU-01, Cat#625902, AB_2210041), Alexa 488 anti-mouse (Cat#A28175, AB_2536161) and Alexa 555 anti-rabbit (Cat#A27039, AB_2536100) from Thermo Fisher (Pittsburg, PA), and horse radish peroxidase-conjugated goat anti-mouse (Cat#170–6516, AB_11125547) and goat anti-rabbit (Cat#170–6515, AB_11125142) antibodies from Sigma-Aldrich (St. Louis, MO).

## Primary screen

HEK cells stably expressing the T2 sensor (containing ANGPTL3 linker sequence with $T_{225}G$ modification (*Song et al., 2014*)) or the T3 sensor (containing FGF23-based linker [*Song et al., 2014*]) were cultured in MEM (Corning, NY, Cat#10–010-CV) with 10% fetal bovine serum (FBS, Atlanta Biologicals, Flowery Branch, GA, Cat#S111150) and 100 IU/ml penicillin-streptomycin (Sigma-Aldrich, Cat#P4333) at 37°C, 5% $CO_2$. Positive controls were cell lines expressing matched sensors with the glycosylation site mutated (Δglycan), specifically $T_{225}G/T_{226}G$ and $T_{178}G$ for the T2 and T3 sensors, respectively (*Song et al., 2014*). Cells (50,000/well) were seeded in flat bottom 96-well-plates (Corning, NY, Cat#3997) and grown for 24 hr. Compounds (the diversity set from ChemBridge Corporation [Chicago, IL] and the approved oncology drugs set V and the diversity set II from the National Cancer Institute Developmental Therapeutics Program) were then added to achieve a 10 µM final concentration. After 6 hr, the medium was aspirated and the cells were released by adding 100 µl 5 mM EDTA/PBS containing 110 nM MG dye (Sharp Edge Labs, Pittsburgh, PA) for 5 min at 37°C. The plates were then transferred to an Accuri C6 flow cytometer (BD Biosciences) where GFP and MG fluorescence was measured using 488 nm and 640 nm for 10,000 cells per well. Data analysis used FlowJo software (www.flowjo.com, SCR_008520). For each well the geometric means of the MG and GFP signals were used to compute the MG/GFP ratio. Each compound was analyzed in two wells and the average of the two resulting ratios (R) was recorded. Each daily run included at least 16 wells of vehicle-only controls (sensor-expressing cells treated with a matching DMSO concentration [Fisher Scientific, Cat#BP231-100]) and a similar analysis was used to calculate their average MG/GFP ratio ($R_{Neg}$) and its standard deviation ($SD_{Neg}$). The Q value of each compound (as well as the untreated Δglycan positive controls) was calculated by using the following equation: $Q=(R-R_{Neg})/SD_{Neg}$. Background fell within the range $-2.5 \leq Q \leq 3$ and the average $Q_{\Delta glycan}$ was 135 and 38 for the T2 and T3 sensors, respectively.

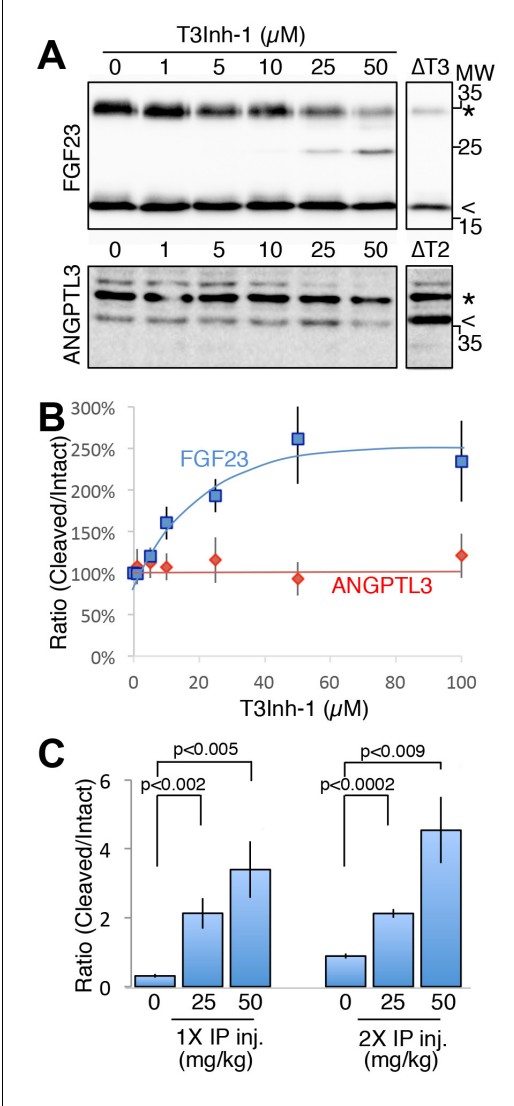

**Figure 4.** T3Inh-1 increases cleavage of FGF23. (**A**) Immunoblot of media collected from cells after a 6 hr period in the presence of the indicated concentrations of T3Inh-1. HEK cells were transfected with FLAG-FGF23 and ppGalNAc-T3 or Myc-ANGPTL3 and anti-FLAG and anti-Myc antibodies were used to assay intact (*) and cleaved (<) FGF23 and ANGPTL3, respectively. The identity and origin of the unmarked band (at approximately 25kD) is unknown and its presence was variable. (**B**) Quantified results showing the percent ratio change of cleaved/intact FGF23 or ANGPTL3 normalized to the amount present in untreated controls (n ≥ 3 ± SEM). (**C**) Serum ELISA assay results showing ratio of cleaved/intact FGF23 in mouse sera collected 24 hr after either 1 or 2 (consecutive day) intraperitoneal injections of the indicated amount of T3Inh-1 (averages of 4 animals ±SEM). P-values are from two-tailed Student's t test.

*Figure 4 continued on next page*

## Secondary screen

Glycosylation assays using recombinant ppGal-NAc-T2 and ppGalNAc-T3 were carried out using the UDP-Glo Glycosyltransferase assay kit (Promega, Madison, WI, Cat#V6962), according to the manufacturer's recommendation. The reaction (25 µl) included 2.5 ng/µl purified enzyme, 25 µM UDP-GalNAc (Sigma-Aldrich, Cat#U5252), 12.5 µM EA2 peptide (AnaSpec, Mucin 10, AA153-165, PTTDSTTPAPTTK, Cat#AS-63841), 25 mM Tris-HCl (pH7.5) (Fisher Scientific, Cat#77-86-1), 5 mM $MnCl_2$ (Fisher Scientific, Cat#M87-500), 2.5 mM $CaCl_2$ (Fisher Scientific, Cat#C70-500) and 50 µM compound. The negative control was vehicle only (same reaction mixture with a matched percentage of DMSO instead of compound), whereas background was from the reaction carried out without enzyme and DMSO instead of compound. All reactions were incubated at 37°C in a water bath for 30 min and then cooled to room temperature. Aliquots (5 µl) were then added to a 384-well plate (Thermo Scientific, Waltham, MA, Cat#164610) to which 5 µl of UDP Detection Reagent was also added. Duplicate measures were made for all reactions. After 1 hr at room temperature the luminescent signals were determined using a Tecan Infinite M1000 (Tecan Group Ltd., Männedorf, Switzerland) with integration time set to 1000 ms. The background-subtracted average for each compound was expressed as a percentage of the negative control (taken as 100%). Compounds with ≥50% effect were considered direct modulators.

## Titration assays

For the sensor assay, about 200,000 cells expressing either sensor were seeded into Greiner Bio-One 24-well plates (Sigma-Aldrich, Cat#662160). After 24 hr, the cells were incubated for another 6 hr in the presence of 0–50 µM compound. To release the cells the medium was replaced with 200 µl 5 mM EDTA/PBS containing 110 nM MG dye. After 5 min at 37°C, fluorescence measurements (20,000 cells/well) were carried out as described above. For the biochemical assay, the assay conditions were identical except for variations in the compound or substrate concentrations as indicated in the figure legends. Data analysis was using Prism (Graph Pad Prism Inc., SCR_002798).

*Figure 4 continued*

The following source data and figure supplements are available for figure 4:

**Source data 1.** Panel B: Cleaved/intact FGF23 and ANGPTL3 secreted at differing T3Inh-1 concentrations.

**Source data 2.** Panel C: Cleaved/intact FGF23 in mouse serum after one two T3Inh-1 injections.

**Figure supplement 1.** Secreted and cellular FGF23 after T3Inh-1 treatment.

**Figure supplement 1—source data 1.** Cleaved versus intact FGF23 secreted at differing T3Inh-1 concentrations.

## Tryptophan fluorescence quenching assay

The purified lumenal domain of ppGalNAc-T3 (30 ng/ μl) was incubated with 0–500 μM compound at room temperature for 10 min and 200 μl aliquots were transferred to a Greiner Bio-One 96-well glass-bottom plate (Sigma-Aldrich, Cat#655892) and the fluorescent emission was scanned (300–450 nm) using a Tecan Infinite M1000 with excitation at 290 nm, gain set to 150, number of flashes at 50 and flash frequency at 400 Hz. The value at the peak of emission at 324 nm was used for the binding curve analysis by Prism.

## Microscopy

For determination of sensor activation, spinning-disk confocal microscopy was used exactly as described (*Song et al., 2014*). To assess possible effects of compounds on Golgi markers, including ppGalNAc-T2 and ppGalNAc-T3, immunofluorescence was carried on HeLa cells treated with 10 μM compound for 6 hr. Briefly, the cells were grown on 12 mm diameter coverslips (Fisher Scientific, Cat#12-545-81) for 48 hr, treated with the compounds, washed with PBS and fixed with 3% paraformaldehyde (Sigma-Aldrich, Cat#P6148) for 15 min. Blocking, Triton X-100 permeabilization, antibody incubations and image capture by spinning-disk confocal were as described (*Mukhopadhyay et al., 2010*). Monoclonal antibodies against ppGalNAc-T2 and T3 were used undiluted and the polyclonal against GPP130 was used at 1:2000. All corresponding images were acquired and adjusted using identical parameters.

## Lectin staining

HeLa cells (treated with T3Inh-1 for 24 hr at the indicated concentrations) were washed twice in PBS containing 0.5% FBS then stained for 30 min with fluorescent lectins ConA (Cat#FL-1001), WGA (Cat#FL-1021), SNA (Cat#FL-1301) or VVA (Cat#FL-1231) (Vector Laboratories, Burlingame, CA) at 1:100 dilutions (except WGA used at 1:1000) in the wash buffer. After staining, the cells were washed twice, lysed with 0.2% Triton X-100 (Fisher Scientific, Cat#BP151-100) for 15 min at 4°C, centrifuged at 14,000 x g for 15 min at 4°C, and the supernatants were read in a Greiner Bio-One 96-well glass-bottom plate using a Tecan Infinite M1000 with excitation at 495 nm and emission from 510–550 nm. Three independent trials were carried out and the peak value (520 nm) was used for quantification.

## Proliferation assay

Equal numbers of HEK or MDA-MB231 cells were plated in Greiner Bio-One 24-well dishes in growth medium containing 0–50 μM T3Inh-1. After 24, 48 or 72 hr, the cells were released using trypsin and counted twice using a hemocytometer for three independent trials.

## Cell invasion assay

Breast cancer MDA-MB231 cells were grown in DMEM medium (Corning, NY, Cat#10–013-CV) with 10% FBS and 100 IU/ml penicillin-streptomycin at 37°C and 5% $CO_2$ and then plated at a density of $1.32 \times 10^4$ in 0.3 ml of DMEM medium without FBS into the upper chamber of a Bioboat insert fitted with a 8.0 μm PET membrane (Corning, NY, Cat#354578). For migration assays, the filter was uncoated. For invasion assays, it was pre-coated with 100 μl Matrigel (BD Biosciences, Cat#356234) at concentration of 272 μg /ml for 1–2 hr at room temperature. Medium containing 10% FBS (0.6 ml) was placed into the lower chamber as a chemo attractant. The compound (final concentration of 5 μM) or a matching amount of DMSO was added to both chambers. After 24 hr or 48 hr the cells were fixed with −20°C methanol for 15 min and then stained with Trypan blue for 5 min. Cells on the upper surface were removed using cotton swabs. Cells present on the underside of the

membrane were photographed using an EVOS FL Cell Imaging System (Invitrogen, CA) and the images were thresholded for presentation and counting using Image J (National Institutes of Health, Bethesda, MD, SCR_003070). The assays involving MCF7 cells were identical except that they were performed 24 hr post transfection.

## Immunoblotting

The FLAG-tagged FGF23, Myc-tagged ANGPTL3 with the $T_{225}G$ modification (*Song et al., 2014*), and untagged ppGalNAc-T3 (from [*Kato et al., 2006*] and cloned into PCDNA 3.0 using BamH1 sites) were transfected into HEK cells using the JetPEI transfection reagent (VWR International, Radnor, PA, Cat#101–40N) according to the manufacturer's instructions. After 24 hr, the medium was replaced with serum-free MEM containing the compound for 6 hr. The medium and cells were then collected and, after trichloroacetic acid precipitation of the medium, analyzed by immunoblot using anti-FLAG antibody at 1:1000 or anti-Myc antibody at 1:2000 and then the peroxidase-coupled secondary antibodies. Emission was captured and quantified using a ChemiDoc Touch Imaging System with Image Lab Software (BioRad, SCR_014210). For ppGalNAc-T3 determinations in different cell lines, cells were collected and lysed with 100 µl buffer (10 mM Tris-HCl (pH8.0), 1 mM EDTA (ACROS ORGANICS, Cat#446085000), 1% Triton X-100, 0.1% sodium deoxycholate (Fisher Scientific, Cat#BP349-100), 0.1% SDS (Fisher Scientific, Cat#BP166-500), 140 mM NaCl (Fisher Scientific, Cat#S271-3), 1 mM PMSF (Sigma-Aldrich, Cat#PMSF-RO). Then 15 µl of each lysate was analyzed by immunoblotting using the purchased anti-ppGalNAc-T3 and anti-α-tubulin antibodies.

## Animal analysis

Wild-type C57BL/6 six to eight week old mice were purchased from Charles River Laboratories international Inc. (Wilmington, MA). Protocols, handling, and care of the mice conformed to protocols approved by the Institutional Animal Care and Use Committee of Carnegie Mellon University (CMU IACUC protocol AS16-005). The compound was dissolved in DMSO at 25 and 50 mg/ml then further diluted with PEG400 (Hampton Research, CA, USA, HR2-603) to create 5 and 10 mg/ml stocks for injection. Control (vehicle only: 20% DMSO, 80% PEG400) and experimental (25 or 50 mg/kg compound) animals received either single or double (separated by 24 hr) intraperitoneal injections and, 24 hr after the last injection, a cardiac blood draw was carried out. The cleaved/intact FGF23 ratio was determined using ELISA kits from Immunotopics (Carlsbad, VA, USA, Cat#60–6800, Cat#60–6300) with cleaved equaling total minus intact.

## Acknowledgements

We thank Dr. Henrik Clausen (University of Copenhagen) for contribution of essential reagents and advice with the manuscript, HaiBing Teng for help with the imaging, Yehuda Creeger for help of flow cytometry, Collin Bachert and Dr. Lingtao Jin for technical and editorial help and Dr. Tina Lee, Dr. Somshuvra Mukhopdhyay and Emily Simon for helpful manuscript suggestions. Funding was from NIH grant 1R21DE026714 to ADL.

## Additional information

### Funding

| Funder | Grant reference number | Author |
| --- | --- | --- |
| National Institutes of Health | 1R21DE026714 | Adam Linstedt |

The funders had no role in study design, data collection and interpretation, or the decision to submit the work for publication.

### Author contributions

LS, Conceptualization, Data curation, Formal analysis, Writing—original draft; ADL, Conceptualization, Supervision, Funding acquisition, Methodology, Project administration, Writing—review and editing

**Author ORCIDs**
Adam D Linstedt, http://orcid.org/0000-0003-0754-1638

**Ethics**
Animal experimentation: Protocols, handling, and care of the mice conformed to protocols approved by the Institutional Animal Care and Use Committee of Carnegie Mellon University. (CMU IACUC protocol AS16-005).

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
