## [Decision Letter]

Thank you for submitting your article "Inhibitor of GalNAc-T3-mediated O-glycosylation blocks cancer cell invasiveness and lowers FGF23 levels" for consideration by *eLife*. Your article has been reviewed by two peer reviewers, one of whom, Christopher G Burd (Reviewer #1), is a member of our Board of Reviewing Editors, and the evaluation has been overseen by Vivek Malhotra as the Senior Editor.

The reviewers have discussed the reviews with one another and the Reviewing Editor has drafted this decision to help you prepare a revised submission.

Summary:

Song and Linstedt report the results of a small molecule screen designed to identify modifiers of the GalNAc-T2 and GalNAc-T3 GalNAc-transferases. The screen implemented clever, previously published and validated, cell-based assays that report GalNAc-T2 and GalNAc-T3 dependent modifications that generate fluorescent signals. A compound that inhibits GalNAc-T3, termed "T3Inh-1", was identified and characterized. Evidence is presented that T3Inh-1 inhibits GalNAc-T3 activity by binding to the lumenal portion of the enzyme. Assays of migration and invasion using cultured MDA-MB231 cells indicate that T3Inh-1 inhibits these properties of the cells.

Essential revisions:

1) The conclusion regarding the specificity of T3Inh-1 for Galnt3 is not adequately supported. The effect of T3Inh-1 on the in vitro activity of purified Galnt6, which is more closely related to Galnt3 than to Galnt2 (see: Revoredo et al., 2016), should be included in the manuscript in order to more thoroughly validate the specificity of T3Inh-1.

2) The cell-based fluorescence lectin binding data does not adequately support the authors' conclusion that T3Inh-1 has no significant effects on synthesis of the major classes of glycans. For example, GSII is not appropriate since few complex N-glycans terminate with GlcNAc. Lectins that detect terminal Gal or sialic acid and differential N-glycan branching, as well as missing O-GalNAc glycans should be investigated. A broader range of lectins needs to be implemented and the cells should be exposed to increasing concentrations of T3Inh-1 for at least 24 and 48 hours. It is recommended that changes in glycosylation should be determined by a more quantitative approach than that presented in the manuscript, such as quantitative flow cytometry.

3) Regarding the effects of T3Inh-1 of FGF23 cleavage and signaling, a concern lies with the presentation and interpretation of the FGF23 processing assays shown in Figure 4. As the authors point out that loss of modification of FGF23 by GalNAc-T3 should result in a T3Inh-1 dose-dependent increase in the amount of cleaved product. The data that tests this, however, show a dose-dependent decrease in the amounts of the intact protein, but the amounts of the cleaved product remain essentially constant (or perhaps it decreases a bit?). The authors need to explain how these data support their conclusion that T3Inh-1 increases cleavage of FGF23, rather than elicits some other outcome, such as failure to secrete the intact protein and/or degradation of the intact protein. The authors state, but do not show, that FGF23 levels in cell lysates were unaffected by T3Inh-1; the reviewers feel these data should be shown because it might be insightful for determining how T3Inh-1 causes the observed change in the ratio of cleaved/uncleaved protein. Perhaps the unglycosylated, intact form is retained and/or degraded within the cell, but the cleaved form is not. Better yet would be a pulse-chase assay of radiolabeled FGF23, which would directly test the authors' hypothesis that the intact form is unstable/degraded in the culture medium. Related, the authors interpret change in the ratio of cleaved/uncleaved protein to reflect a net change in "signaling activity" (Results section), but no assays of signaling activity are reported. The text should be modified to accurately reflect this.

[Editors' note: further revisions were requested prior to acceptance, as described below.]

Thank you for resubmitting your work entitled "Inhibitor of GalNAc-T3-mediated O-glycosylation blocks cancer cell invasiveness and lowers FGF23 levels" for further consideration at *eLife*. Your revised article has been favorably evaluated by Vivek Malhotra (Senior editor), a Reviewing editor and one reviewer.

The manuscript has been improved but there are some remaining issues that need to be addressed before acceptance, as outlined below. All of these points can be addressed by revisions to the text.

1) The authors must use the established nomenclature throughout the text and in all figures. The enzymes that add GalNAc to protein are called polypeptide GalNAc-transferases. The shorthand for each of the 20 members of the family is ppGalNAcT-1 etc, and the genes are named GALNT1 etc. The leaving out of polypeptide by the authors confuses the enzymes under study with GalNAcTs which transfer GalNAc to other sugars, a totally distinct class of enzyme.

2) Regarding the new lectin binding assay, the authors should qualify their conclusions by stating that no major change in the binding of lectins that detect N- or O-glycans was induced by inhibitor treatment. The lectin binding assay, though acceptable, is not standard practice for the field (flow cytometry is). Additionally, the sensitivity of the fluorescence assay is not clear since there are no positive controls with altered N- or O-glycans. There are many other types of glycans that would not be detected by the lectins used in their assay. The amount of time that cells were incubated with inhibitor before lectin binding was assayed must be indicated.

3) The authors should explain the origin of the band at ~25 kD in Figure 4. This gel does not show a visible increase in the ~15 kD cleavage product which is very abundant at time zero. More explanation of these data is warranted.

4) The Materials and methods should include precise details regarding origin, clone name, catalogue number and dilution used for each antibody or reagent and precise conditions of use in the relevant experiment. For example the source of GPP130 is given as the reference for Puri, Bachert, Fimmel and Linstedt, 2002, which is not sufficient. Enough information should be provided so that precise replication of each experiment is possible.

---

## [Author Response]

*Essential revisions:*

*1) The conclusion regarding the specificity of T3Inh-1 for Galnt3 is not adequately supported. The effect of T3Inh-1 on the* in vitro *activity of purified Galnt6, which is more closely related to Galnt3 than to Galnt2 (see: Revoredo et al., 2016), should be included in the manuscript in order to more thoroughly validate the specificity of T3Inh-1.*

As requested, we have performed a test of the inhibitor (T3Inh-1) against GalNAc transferase-6, which according to a phylogenetic analysis is most closely related to GalNAc-T3. These data are now added to Figure 2. Gratifyingly, there was no detectable activity despite the full and expected potency against GalNAc-T3 in parallel assays.

*2) The cell-based fluorescence lectin binding data does not adequately support the authors' conclusion that T3Inh-1 has no significant effects on synthesis of the major classes of glycans. For example, GSII is not appropriate since few complex N-glycans terminate with GlcNAc. Lectins that detect terminal Gal or sialic acid and differential N-glycan branching, as well as missing O-GalNAc glycans should be investigated. A broader range of lectins needs to be implemented and the cells should be exposed to increasing concentrations of T3Inh-1 for at least 24 and 48 hours. It is recommended that changes in glycosylation should be determined by a more quantitative approach than that presented in the manuscript, such as quantitative flow cytometry.*

As requested, we expanded our survey of lectin staining to include Wheat germ agglutinin (WGA) and Sambucus Nigra (SNA), which recognize most major surface glycans. Additionally, the assay was made more rigorously quantifiable by using a plate reader allowing determinations from thousands of cells (n=3 independent trials). These data are present as a new figure: Figure 1—figure supplement 2. There was no significant diminishment of staining for any lectin tested supporting our conclusion that T3Inh-1 is selective against GalNAc-T3 rather than acting as a general inhibitor of glycosylation enzymes.

*3) Regarding the effects of T3Inh-1 of FGF23 cleavage and signaling, a concern lies with the presentation and interpretation of the FGF23 processing assays shown in Figure 4. As the authors point out that loss of modification of FGF23 by GalNAc-T3 should result in a T3Inh-1 dose-dependent increase in the amount of cleaved product. The data that tests this, however, show a dose-dependent decrease in the amounts of the intact protein, but the amounts of the cleaved product remain essentially constant (or perhaps it decreases a bit?). The authors need to explain how these data support their conclusion that T3Inh-1 increases cleavage of FGF23, rather than elicits some other outcome, such as failure to secrete the intact protein and/or degradation of the intact protein. The authors state, but do not show, that FGF23 levels in cell lysates were unaffected by T3Inh-1; the reviewers feel these data should be shown because it might be insightful for determining how T3Inh-1 causes the observed change in the ratio of cleaved/uncleaved protein. Perhaps the unglycosylated, intact form is retained and/or degraded within the cell, but the cleaved form is not. Better yet would be a pulse-chase assay of radiolabeled FGF23, which would directly test the authors' hypothesis that the intact form is unstable/degraded in the culture medium. Related, the authors interpret change in the ratio of cleaved/uncleaved protein to reflect a net change in "signaling activity" (Results section), but no assays of signaling activity are reported. The text should be modified to accurately reflect this.*

Figure 4—figure supplement 1 has been added to address issues concerning the intact and cleaved bands of FGF23. The main question is why the loss of intact FGF23 does not appear to be accompanied by a quantitatively comparable increase in the cleaved fragment. There is previous evidence that the cleaved FGF23 fragment is difficult to recover in cultured cell media samples (Kato et al., 2006). Also, note that even in ∆T3 cells the cleaved fragment is relatively under- represented in the media. It also bears mentioning that we do detect a strong increase in this cleaved FGF23 fragment in the serum of injected mice (Figure 4). Finally, the new figure presents the data more thoroughly and shows that for the quantified data the discrepancy is minor. First we present an additional representative blot from the 3 identical trials (panel A of the figure) to illustrate that there was some variation in our recovery of the cleaved fragment. This blot better represents the slight increase that the quantified data show. This quantified data, showing the increase on average in cleaved FGF23 and the loss of intact, is presented in panel B. Taken together with the data in the main body (Figure 4) this provides a comprehensive representation of the result we obtained. Indeed, in data not shown, the experiment was carried out for an additional cell type and with different parameters and, in each case, yielded the same result: a significant loss of intact and a slight increase of cleaved FGF23. In part C-E, we have now added our data for the cell extracts. Neither the intact nor cleaved FGF23 showed a significant change (C-D). Also, panel E compares cells extracts from untreated and chloroquine-treated cells (to block lysosomal degradation). Chloroquine did not cause any accumulation. All together the data in cells and mice argue against changes in expression, sorting, or degradation and support the conclusion that T3Inh- 1 increases FGF23 cleavage.

[Editors' note: further revisions were requested prior to acceptance, as described below.]

*The manuscript has been improved but there are some remaining issues that need to be addressed before acceptance, as outlined below. All of these points can be addressed by revisions to the text.*

*1) The authors must use the established nomenclature throughout the text and in all figures. The enzymes that add GalNAc to protein are called polypeptide GalNAc-transferases. The shorthand for each of the 20 members of the family is ppGalNAcT-1 etc, and the genes are named GALNT1 etc. The leaving out of polypeptide by the authors confuses the enzymes under study with GalNAcTs which transfer GalNAc to other sugars, a totally distinct class of enzyme.*

The text has been changed accordingly using ppGalNAcT to refer to the enzymes.

*2) Regarding the new lectin binding assay, the authors should qualify their conclusions by stating that no major change in the binding of lectins that detect N- or O-glycans was induced by inhibitor treatment. The lectin binding assay, though acceptable, is not standard practice for the field (flow cytometry is). Additionally, the sensitivity of the fluorescence assay is not clear since there are no positive controls with altered N- or O-glycans. There are many other types of glycans that would not be detected by the lectins used in their assay. The amount of time that cells were incubated with inhibitor before lectin binding was assayed must be indicated.*

The text and methods were revised to state that it was a 24 h treatment (already in figure legend). The text was also revised to make clear that our results speak only to the abundant glycans detected by the lectins we used.

*3) The authors should explain the origin of the band at ~25 kD in Figure 4. This gel does not show a visible increase in the ~15 kD cleavage product which is very abundant at time zero. More explanation of these data is warranted.*

The legend to Figure 4 was revised to indicate that the origin and identity of the 25kD band is unknown and its presence in the independent trials was variable. The identity of intact and cleaved FGF23 (marked on the figure and explained in the legend) is clear in these experiments because of their migration on the gel, their presence only in transfected cells, and their expected behavior in cells lacking expression of ppGalNAc-T3.

*4) The Materials and methods should include precise details regarding origin, clone name, catalogue number and dilution used for each antibody or reagent and precise conditions of use in the relevant experiment. For example the source of GPP130 is given as the reference for Puri, Bachert, Fimmel and Linstedt, 2002, which is not sufficient. Enough information should be provided so that precise replication of each experiment is possible.*

The Materials and methods section was extensively revised by addition of information regarding the origin of the reagents used. To our knowledge all dilutions are specified. The reference for anti-GPP130 is the paper describing the origin and first use of the antibody.